# IMSI—Guidelines for Sperm Quality Assessment

**DOI:** 10.3390/diagnostics12010192

**Published:** 2022-01-13

**Authors:** Krzysztof Lukaszuk, Grzegorz Jakiel, Izabela Wocławek Potocka, Jolanta Kiewisz, Jolanta Olszewska, Wlodzimierz Sieg, Amira Podolak, Ewa Pastuszek, Artur Wdowiak

**Affiliations:** 1Invicta Research and Development Center, 81-740 Sopot, Poland; luka@gumed.edu.pl (K.L.); grzegorz.jakiel1@o2.pl (G.J.); wlodzimierz.sieg@invicta.pl (W.S.); ewa.pastuszek@invicta.pl (E.P.); 2Department of Obstetrics and Gynecological Nursing, Faculty of Health Sciences, Medical University of Gdansk, 80-210 Gdansk, Poland; jolanta.olszewska@gumed.edu.pl; 3iYoni App by LifeBite, 10-763 Olsztyn, Poland; 4Department of Obstetrics and Gynecology, Center of Postgraduate Medical Education, 01-004 Warsaw, Poland; 5Department of Gamete and Embryo Biology, Institute of Animal Reproduction and Food Research, Polish Academy of Sciences, 10-748 Olsztyn, Poland; i.woclawek-potocka@pan.olsztyn.pl; 6Department of Human Histology and Embryology, School of Medicine, Collegium Medicum, University of Warmia and Mazury, 10-082 Olsztyn, Poland; jolanta.kiewisz@uwm.edu.pl; 7Chair of Obstetrics and Gynecology, Faculty of Health Sciences, Medical University of Lublin, 4-6 Staszica St., 20-081 Lublin, Poland; wdowiakartur@gmail.com

**Keywords:** intracytoplasmic sperm injection (ICSI), intracytoplasmic morphologically selected sperm injection (IMSI), sperm quality, optical microscopy

## Abstract

Intracytoplasmic sperm injection (ICSI) is a widely used and accepted treatment of choice for oocyte fertilization. However, the quality of sperm selection depends on the accurate visualization of the morphology, which can be achieved with a high image resolution. We aim to correct the conviction, shown in a myriad of publications, that an ultra-high magnification in the range of 6000×–10,000× can be achieved with an optical microscope. The goal of observing sperm under the microscope is not to simply get a larger image, but rather to obtain more detail—therefore, we indicate that the optical system’s resolution is what should be primarily considered. We provide specific microscope system setup recommendations sufficient for most clinical cases that are based on our experience showing that the optical resolution of 0.5 μm allows appropriate visualization of sperm defects. Last but not least, we suggest that mixed research results regarding the clinical value of IMSI, comparing to ICSI, can stem from a lack of standardization of microscopy techniques used for both ICSI and IMSI.

## 1. Introduction

Since it was first proposed in 1992, intracytoplasmic sperm injection (ICSI) has become the treatment of choice for a large number of patients, not only those with abnormal sperm parameters [1]. Nowadays, approximately two-thirds of fertilization cycles in Europe are performed with the use of ICSI [2]. However, several concerns about the safety of ICSI and its impact on the offspring have been raised [3,4]. These fears arise from the risk of the potential use of an abnormal spermatozoon to be injected into the egg.

Although the clinical significance of the assessment of sperm morphology is still debatable, it has recently been recognized that an accurate measurement of morphological anomalies plays an essential role in the determination of male fertility. In cases of severe oligozoospermia when ICSI appears to be the only chance for the couple, the assessment of sperm morphology seems to play a key role. Efforts to improve the morphological assessment resulted in the development of motile sperm organelle morphology examination (MSOME, using Nomarski differential interference contrast (DIC) microscopy and high magnification), first described by Bartoov and colleagues in 2001 [5]. This technique allows for the better evaluation of a spermatozoon’s morphology and better visualization of sperm head vacuoles. The first study assessed the pregnancy rate in 24 couples that underwent one intracytoplasmic morphologically selected sperm injection (IMSI) cycle. All couples were diagnosed with male infertility and had at least two previously failed ICSI cycles. In each case, a spermatozoon without nuclei abnormalities was selected by MSOME and injected into the egg. The pregnancy rate in the study was 58% [5]. The authors concluded that the morphology of the sperm is a significant factor in embryo quality following ICSI. Subsequent studies presented contradictory results. Some randomized studies and a low powered meta-analysis showed that IMSI procedures yielded significantly higher fertilization rates [6,7,8,9,10], better embryo development [8,9,11,12,13,14,15,16] and higher clinical pregnancy rates [5,7,11,12,13,15,17,18,19,20,21,22] and lower miscarriage rates [13,16,22] or live birth [13,17]. We also found that the results of high-quality MSOME show a correlation with sperm DNA fragmentation [23]. Other authors, however, observed slight or almost no improvements in the clinical outcomes [15,24,25,26,27,28,29,30,31,32].

There is no, for the moment, clear proof that IMSI offers any benefits. Some differences in the results may be due to variations in microscope systems used for ICSI and IMSI, as the improvement in IMSI may depend on the quality of the ICSI system used for comparison. We will show that a high-quality ICSI system may offer almost the same level visualization of sperm morphology as IMSI. Some of the doubts about IMSI are associated with methodological issues originating from fundamental mistakes in describing the equipment used. Misconceptions about the terms that are commonly used when describing microscopy systems also lead to confusion. Resolution can be understood very differently depending on the field. In technology, e.g., when evaluating monitors, it describes the clarity and sharpness of the displayed image, i.e., the display resolution. However, resolution in microscopy refers to optical resolution, which describes the ability to distinguish details with the use of a microscope.

In our opinion, it is high time to resolve these issues with the introduction of the standardization of microscopy techniques.

In various recent, publications, the authors claimed that spermatozoa are observed under 6000×–10,000× magnification [33,34,35,36,37,38]. These assertions are misleading and can prompt an incorrect conviction that this kind of magnification can be achieved with a standard optical microscope (or even a most advanced one). We would like to provide some theoretical background of the physical laws of magnification and resolution to show the scope of the problem. However, the reader should bear in mind that the information below is somewhat simplified and is not addressed to optical microscopy specialists, but instead is meant as a more practical guide to embryologists and physicians.

## 2. Magnification and Resolution

To obtain a microscope’s total magnification, we need to consider the parameters of the eyepieces and the objective lenses. Mainly, we do not seek a larger image, i.e., a magnified image, but rather, we are interested in obtaining more details. Magnification is dependent on the obtained resolution. Optical resolution should be understood as the smallest distance between two points on a specimen that can still be differentiated as two separate objects.

The optical resolution of a light microscope (*d*) can be calculated according to the formula introduced by Ernst Abbe and represents a measure of the image sharpness [39]:$$d=\frac{\lambda }{2\eta sin\alpha }$$
*d =* optical resolution*λ* = wavelength of the illuminating light used*η* = refractive index of the optical medium between the front lens and cover glass*α* = half the opening angle of the objective used.

Ernst Abbe proposed describing objectives using a parameter that helps to judge its resolution, termed the numeric aperture (*NA*).
NA=ηsinα
*NA* = numeric aperture*η* = refractive index of the optical medium between the front lens and cover glass*α* = half the opening angle of the objective used.

The angle (*µ*) increases as the light cones change, which cause increases in the *NA* and a decreased working distance (Figure 1). This means that the optical resolution depends on NA and on the illuminating light’s wavelength that is used for observation. A higher aperture of the objective and the condenser results in better optical resolution.

It is also worth noting that NA > 1.0 is only achievable with immersion oil. The maximum theoretical opening angle of an objective, though not possible in practice, is 90 degrees. The refractive index of air is 1. Therefore, multiplying it by sin 90 (which is 1) cannot give us NA higher than 1. This would only be possible with the refractive index greater than 1, with an immersion medium of water, glycerol or oil. Only then can NA be greater than 1.

Another issue is working with plastic materials (e.g., Petri dishes, Roux bottles), which are thicker than glass coverslips. It is thus essential to have microscope objectives that are corrected for the thickness of the appropriate materials. Microscope objectives are marked as either “-” (not sensitive), 0.17 (appropriate to coverslip thickness of 0.17 mm), or 1 (appropriate to coverslip thickness of 1 mm). It is crucial to be aware of this number to ensure the correct microscope setup with the objective matching thickness of the material.

Another significant factor in the determination of optical resolution is the light wavelength used to observe a specimen. Shorter wavelengths enable achieving a resolution allowing for the detection of more details. Half of the light wavelength of the microscope’s light source is the cut off beneath which an object will not be visible under that microscope. Therefore, under light microscopes that use visible light (400 nm minimum wavelength), it will never be possible to see any object that is smaller than approximately 200 nm.

In addition to Abbe’s resolution criterion (which is stringent), there is also the Rayleigh criterion (commonly used to determine the optical resolution for spectrometers and imaging devices) [40]:$$d=\frac{1.22\lambda }{2NA}$$

The resolving power of a microscope objective is determined by the numerical aperture. However, the total microscope optical resolution is also influenced by the substage condenser’s numerical aperture so that:$$d=\frac{1.22\lambda }{\left(NAobjective+NAcondenser\right)}$$

To summarize, the higher the numerical aperture of the used microscope system and the shorter the wavelength, the better the optical resolution.

## 3. What Is Then the Optical Resolution of the Optical Systems Used for In Vitro Fertilization (Micromanipulation for ICSI)?

The 20× magnification is a standard magnification used to observe spermatozoa, although some clinics use magnifications of 10× or 40×.

Micromanipulation requires the use of an inverted microscope that should have a long or very long working distance. This is possible, but means that the NA of the microscope objective is somewhat sacrificed. This is the reason why the standard NA is 0.25 for a 10× objective, 0.45 for a 20× objective, and 0.55–0.60 for a 40× objective.

In order to apply the standard formula, we also need to know the NA of the condenser. Standard condensers used for Hoffman contrast or other modulation contrasts have an NA in the range of 0.45–0.6.

Light is the part of the electromagnetic radiation spectrum visible to the human eye. The human eye perceives radiation in the range of 400–700 nm (some sources say 380–760 nm) [41]. However, it is most sensitive to light from the middle range of the visible spectrum—the color green, around 550 nm, and this range is typically used for observation. A filter can be used to limit the wavelength of light on the observed specimen and also to improve the image quality and reduce color aberration. The resolution would be significantly higher with a wavelength of approximately 400 nm. However, such filters are not standard equipment of detecting systems.

Let us now calculate the typical optical resolution of an optical system used in an in vitro fertilization procedure.

Standard ICSI system consists of a 550 nm filter (or not), a condenser with an NA of 0.3–0.4 and a 20× objective with and NA of 0.4. The optical resolution will be from 0.84 to 1.22 μm.

A well-optimized ICSI workstation would consist of much better optical components. For example, we can assume a standard 550 nm filter and a condenser with an NA of 0.6. The optical resolution for a 20× objective with an NA = 0.45 would be 0.64 μm (with a 400 nm filter, it would be 0.47 μm). In the case of a 40× objective with an NA = 0.6, the optical resolution would be 0.56 μm (with a 400 nm filter, it would be 0.41 μm).

## 4. What Is Then the Optical Resolution of the Systems Used for IMSI?

Generally, there are four commercially available IMSI systems. Two of them depend on optical techniques, and two depend on working with or without immersion oil.

For IMSI (and also ICSI) and standard observation in in vitro fertilization (IVF), we usually use different types of modulation systems. The first technique, Hoffman modulation contrast (HMC) microscopy, was invented by Robert Hoffman in 1975 [42,43]. It enhances the contrast in unstained biological samples by using special components (a rotating polarizer, condenser slit plate and modulator) in the light paths. HMC microscopy relies upon the use of sample phase gradients. The light refracting path shifts somewhat wherever a break appears due to a rapid spatial change in the optical path passing through the sample. A pseudo-relief image is obtained as the light passes through the modulator and is underlined in contrast. The three dimensions represent light phase gradients, not the actual geometry of the object under observation. As a result, HMC can be used with birefringent samples that are not suitable for DIC analysis (e.g., samples placed in plastic Petri dishes). This system is the gold standard, providing cheap and easy-to-use visualizations of unstained cells and tissues. After the patent expired, other manufactures copied this technique with greater and lesser success. These led to the availability of systems with different names and varying quality—ZEISS plasDIC and improved Hoffman modulation contrast (iHMC), Nikon advanced modulation contrast (NAMC) and emboss contrast, Olympus Relief contrast (RC) and Leica’s integrated Hoffman modulation contrast (iMC). These systems are relatively cheap, and importantly in commercial applications, they are characterized by low operating costs. Moreover, they can be used with plastic components. Thus, this allows us to use this standard equipment for our observation and intervention procedures.

The second type of technique is DIC microscopy, also known as Nomarski interference contrast (NIC), developed by Jerzy Nomarski, a Polish physicist, in 1952 [44,45]. It was also invented to enhance the contrast in undyed, transparent samples. This type of system is much more complex and challenging to set up and maintain. In DIC imaging, the process begins by passing the light through a polarizing filter and splitting it into two rays that are polarized to each other at an angle of 90°. As they pass through adjacent areas of the sample, they are separated by shearing. The second prism recombines them, which leads to interference. The two combined rays pass through the sample at adjacent points and have, therefore, slightly different phases. This phase difference is due to the difference in the length of the optical path, and the process of recombination creates the resulting image that enhances contrast in unstained, transparent samples. DIC also requires the use of glass components in the optical pathway. We are unable to use plastic dishes, which are the standard dishes for IVF procedures.

The second decision to optimize our IMSI system is whether to use immersion. This affects the setup, operational and maintenance costs and time consumption.

The maximal optical resolution which we can achieve in the systems without immersion is between 0.27 μm (with NA of both condenser and objective at 0.9 and filter 400 nm), for interventional systems (micromanipulation, biopsy, etc.) and 0.45 μm (objective NA 0.9 and condenser NA 0.72 and filter 550 nm).

If we choose a system with immersion, we can achieve an optical resolution for the observational setup of 0.18 μm (objective NA 1.4 and condenser NA 1.25 and filter 400 nm), and a resolution of 0.32 μm for the interventional setup (objective NA 1.4 and condenser NA 0.72 and filter 550 nm).

## 5. What Are the Paremeters of the System Sufficient for ICSI and IMSI?

We can proceed to design an optimal system used without immersion, while implementing the least expensive components to obtain a satisfactory optical resolution. We assume that an optical resolution of 0.5 μm would be satisfactory for most clinical cases and allow the visualization of sperm defects [46].

If the goal is to make sperm defects visible, what kind of system should be built to carry out the procedure?

It appears that we should be able to achieve the expected result with the use of a good-quality objective with 40× magnification. This should be possible with a microscope objective with an NA of 0.6. There are also available systems with modulation contrast for 63× magnification and an NA of 0.8 and 60× magnification with an NA of 0.7 (for a 400 nm filter and an NA 0.6 condenser, with an optical resolution of 0.38 μm).

Why should we consider systems based on HMC? These systems are significantly cheaper than those based on DIC. Additionally and importantly, the cost of consumables used is much lower. The DIC system does not allow for the use of plastic culture dishes. Systems based on contrasts modulation do not require the use of specialized components nor frequent calibration by microscope service companies.

Assuming the following parameters:▪the blue light filter of 400 nm▪condenser with working distance (WD) = 40 mm and NA 0.6▪63× or 100× objective with NA 0.9
we can achieve an optical resolution of 0.3 μm. We should note that if we were to use a 550 nm filter, we would only get an optical resolution of 0.4 μm.

Using a good-quality objective with an NA of 0.6 (for example, 40×), we can achieve an optical resolution of 0.41 and 0.56 μm, respectively.

In the Appendix A, we present photos illustrating the differences between various techniques, magnifications and NA (Figure A1, Figure A2, Figure A3 and Figure A4) and a table showing values of recommended magnification (Table A1).

## 6. Do We Need a Camera for Better Visualization of the Spermatozoa Morphology?

It is well established that the standard resolution of the human eye is 300 pixels per inch [47]. This means that when you are looking at a well-illuminated object from a distance of 30 cm, it gives a resolution of 12 pixels per 1 mm—thus, we can see pixels of a diameter of 0.08 mm, or 83 μm.

The question is, how much do we have to magnify the picture to see the maximum available resolution of our system?

As previously mentioned, for a standard ICSI system, we have an optical resolution of 0.84 μm, for high-quality ICSI systems, we have an optical resolution of 0.41 μm, for IMSI without immersion, we have an optical resolution of 0.33 μm, and for IMSI with immersion, we have an optical resolution of 0.18 μm. Thus, to make a 0.08 mm pixel visible, we need to enlarge it, so we have to magnify the picture 100× (objective 20× NA 0.4), 202× (objective 40× NA 0.6), 251× (objective 63× NA 0.9) and 461× (objective 100× NA 1.4), respectively, for each system type.

Because in most cases we use 10× magnifying oculars, we use magnifications of 200×, 400×, 630× and 1000×, which do not require any additional magnification to attain everything we can get from our microscopes. Applying digital magnification is not necessary, as it does not enable us to see additional details.

Aging often means a worsening eyesight, and the visible resolution can reduce by as much as three times (down to four pixels per 1 mm). In such a case, this would mean magnification of 300×, 606×, 753× and 1383×, respectively. However, this can be sufficiently overcome by using a 1.5× magnifier, which is the standard accessory of most advanced microscopes. The same effect could be achieved using a camera and on-screen magnification, but this would cause an additional technical inconvenience—the monitor would have to be used for the selection of spermatozoa.

## 7. Our Experience

For this publication, we checked most of the systems we have in our clinics. We use the following microscopes: Leica—DMIL6000, Nikon TE2000, Olympus IX71 and Zeiss Axiovert 200.

We evaluated microscope objectives with 20×, 40×, 60–63×, and 100× magnification, with different apertures, using different optical configurations and techniques.

Our subjective impression is that the best clinical choice is the Nikon system, where we can obtain 2.5× magnification in the ocular column, allowing 1000× magnification with a 0.4 μm optical resolution in the ocular column without any additional equipment or sophisticated solutions with a 2.7–3.7 mm working distance.

Our choice for research is the Leica DMI6000, which allows for fast and easy changes of microscopic techniques and is very resistant to any personnel-dependent problems.

For routine work when IMSI not needed often, we found Olympus systems to be most user-friendly.

## 8. Conclusions

There is still no consensus regarding the clinical value of IMSI. The contradictory results are caused by the lack of standardization of microscopy techniques used for ICSI and IMSI.

Clinical use of IMSI should not require attempts to achieve the maximum optical resolution of sperm images, especially due to the fact that this often causes additional technical difficulties, prolongs the procedures, and affects the homeostasis of the gametes. Good contrast on transparent specimens can be achieved more simply and less expensively with HMC rather than DIC. An additional benefit of using HMC is the compatibility with birefringent plates (such as plastic Petri dishes). The DIC will not work correctly in those conditions. We can assume that an optical resolution of 0.5 μm is sufficient for the selection of a spermatozoon for fertilization. In a properly configured ICSI system, we are able to obtain the optical resolution of 0.4 μm. Dedicated IMSI systems can achieve an optical resolution of 0.33 μm without immersion oil and 0.18 μm while using immersion oil. Neither one is necessary to properly conduct the procedure, and both are related to the above-mentioned risks. The use of cameras to obtain additional magnification does not improve quality when using 1.5× magnification microscopes or 15× eyepieces.

Comparisons between IMSI and ICSI fertilization cycles may yield different results depending on the microscope setups used for ICSI and IMSI. When a high-quality system is used for ICSI that allows for visualizations of sperm vacuoles and obtained results are compared with IMSI, there may not be an apparent improvement. However, the clinics with a less advanced ICSI setup may experience improvements with the use of IMSI. This is why it is crucial to know what is being compared. In order to assess the influence of sperm vacuoles, the first step should be to set up the ICSI and IMSI systems and determine the size of vacuoles that can be visualized with each system. Once it is established what kind of vacuoles are visible, the evaluation of the differences between ICSI and IMSI and the influence of sperm vacuoles of different sizes can be accomplished.

To decide whether you need the improvement, you can evaluate your microscope system using the App “Resolution” for Android devices (downloadable from the Google Play Store at https://play.google.com/store/apps/details?id=com.Barlowax.resolutionfragments&hl=en, accessed on 25 November 2021) or the iOS equivalent “Microscope Resolution” (downloadable from the App Store at https://apps.apple.com/us/app/microscope-resolution/id1229786939, accessed on 25 November 2021).

The foregoing conversions show that sometimes small and inexpensive adjustments to the system in use can significantly improve the quality of the performed procedures.

Moreover, the authors of publications in the area of assisted reproduction should describe not only the used microscope magnification, but most of all the optical resolution at which the image was obtained (or all optical components of the system used for micromanipulation). This would allow for the correct and reliable interpretation of the results that is possible only if it is accurately reported as to what sperm defects were visualized. In our opinion, the lack of this information is the reason why the assessment of the clinical value of IMSI still requires more research.

## Figures and Tables

**Figure 1 diagnostics-12-00192-f001:**
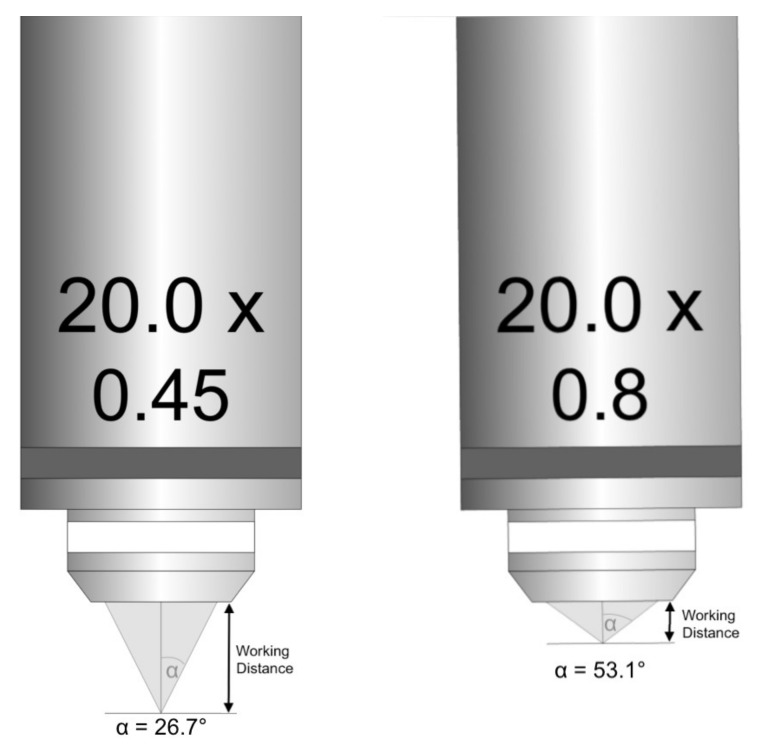
Inverse relationship of numerical aperture and working distance.

## Data Availability

Not applicable.

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
