# Peer review of "IMSI—Guidelines for Sperm Quality Assessment"

_diagnostics, 2022, doi:10.3390/diagnostics12010192_

Round 1

Reviewer 1 Report

Dear Authors,

This manuscript is very important for IVF Units.

However, some important points should first be addressed:

-There are a few English errors.

-Authors should avoid terms where they seem to criticize readers. Instead, use only teaching information.

-What we, embryologists, really need from this manuscript:

Images from Hoffman, 20x and 40x, with the best NA for objectives and condenser, taking into account the 10x of the eyepieces (oculars). In this way, we can confirm what we are using, or what we should ask to our microscope technician to improve.

The way it is presented does not help IVF Units.

Author Response

Response to the Editor and  Reviewers’ comments

Ms. Ref. No.: diagnostics-1506481

Title: “IMSI – Guidelines for sperm quality assessment”

Journal: Diagnostics

General comments:

We would like to thank the Co-Editors-in-Chief and Reviewers for their comments and insights. We appreciate your input and have revised the manuscript in line with your suggestions. We hope that the revised version will meet with your approval. Below are our responses to all the Reviewers' comments.

All changes are marked up using the “Track Changes” function.

Answer to the Reviewer Comments:

Comment#1: There are a few English errors.

Authors Response: To meet the Reviewer expectations, the entire manuscript has also been reviewed by a native English speaker and additional corrections were made.

Comment#2: Authors should avoid terms where they seem to criticize readers. Instead, use only teaching information.

Authors Response: Thank you very much for this comment. Our intention was not to criticize or discriminate anyone. However, after the thoughtful text analysis we agree that some phrases may be interpreted this way. Thus, we have revised the manuscript to enhance its quality of the tone and language. All the phrases seem to criticize either authors or readers were replaced with more informative expressions.

Comment#3: What we, embryologists, really need from this manuscript:

Images from Hoffman, 20x and 40x, with the best NA for objectives and condenser, taking into account the 10x of the eyepieces (oculars). In this way, we can confirm what we are using, or what we should ask to our microscope technician to improve.

The way it is presented does not help IVF Units.

Authors Response: Thank you very much for this remark. To meet the Reviewers expectations values of total magnification have been included in the description of each image in the manuscript, thus, they could be used as guidelines for embryologists and their technicians. Additionaly, we decided to provide the table showing recommended values of magnification for the defined parameters as wavelength, condenser NA, objective NA and resolution (including resolution of eyesight). We hope that the revised version will meet with your approval. Thank you for your consideration.

Reviewer 2 Report

The guidelines manuscript titled “IMSI – Guidelines for sperm quality assessment” is well structured and gives important information regarding different parameters for assessing sperm quality by optical microscopy. It is also well written, nonetheless the tone and language used is discriminating, aggressive and generalizes claims of lack of knowledge to the community, for these reason, the manuscript is accepted with mayor corrections. It is advisable that some phrases are changed:

L16-17: These claims, shown in a myriad of publications, demonstrate a lack of basic knowledge of optics not only among authors, but also reviewers and editors in our community.

L69-70: Data presented in our field's publications risks ridiculed by anyone with basic knowledge of optical microscopy.

L74-75: To demystify this erroneous illusion of those incredible magnifications, let's start with the fundamentals.

L138, L231: the admiration mark after seems unnecessary, if considered necessary, would be better to explain your meaning.

L255: Why mention retired embryologists, if they are already retired, it is no longer important if they need a higher magnification in order to perform a job they are already retired from.

In my opinion, these phrases should be reconsidered or, if necessary for the manuscript, rewritten to be less generalized and discriminatory, also use a less aggressive language (i.e. ridiculed, demystify erroneous illusions, lack of basic knowledge of the community).

These discriminatory claims elicited ethical concerns that greatly diminish the quality of presentation and overall merit of the manuscript.

Author Response

Response to the Editor and  Reviewers’ comments

Ms. Ref. No.: diagnostics-1506481

Title: “IMSI – Guidelines for sperm quality assessment”

Journal: Diagnostics

General comments:

We would like to thank the Co-Editors-in-Chief and Reviewers for their comments and insights. We appreciate your input and have revised the manuscript in line with your suggestions. We hope that the revised version will meet with your approval. Below are our responses to all the Reviewers' comments.

All changes are marked up using the “Track Changes” function.

Answer to the Reviewer Comments:

Comment#1: The guidelines manuscript titled “IMSI – Guidelines for sperm quality assessment” is well structured and gives important information regarding different parameters for assessing sperm quality by optical microscopy. It is also well written, nonetheless the tone and language used is discriminating, aggressive and generalizes claims of lack of knowledge to the community, for these reason, the manuscript is accepted with mayor corrections. It is advisable that some phrases are changed:

L16-17: These claims, shown in a myriad of publications, demonstrate a lack of basic knowledge of optics not only among authors, but also reviewers and editors in our community.

L69-70: Data presented in our field's publications risks ridiculed by anyone with basic knowledge of optical microscopy.

L74-75: To demystify this erroneous illusion of those incredible magnifications, let's start with the fundamentals.

L138, L231: the admiration mark after seems unnecessary, if considered necessary, would be better to explain your meaning.

L255: Why mention retired embryologists, if they are already retired, it is no longer important if they need a higher magnification in order to perform a job they are already retired from.

In my opinion, these phrases should be reconsidered or, if necessary for the manuscript, rewritten to be less generalized and discriminatory, also use a less aggressive language (i.e. ridiculed, demystify erroneous illusions, lack of basic knowledge of the community).

These discriminatory claims elicited ethical concerns that greatly diminish the quality of presentation and overall merit of the manuscript.

Authors Response: Thank you very much for this comment. Our intention was not to criticize or discriminate anyone. However, after the thoughtful text analysis we agree that some phrases may be interpreted this way. Thus, we have revised the manuscript rewriting all the phrases mentioned above to enhance its quality of the tone and language.

In terms of retired embryologist, we would like to explain that we had on mind older embryologists who still perform their job. After considering this particular comment we decided to remove the word “retired” to make our thoughts clearer.

Thank you for your consideration. We hope that the revised version will meet with your approval.